# Precision Through Detail: Radiomics and Windowing Techniques as Key for Detecting Dens Axis Fractures in CT Scans

**DOI:** 10.3390/diagnostics15202599

**Published:** 2025-10-15

**Authors:** Karl Ludger Radke, Anja Müller-Lutz, Daniel B. Abrar, Marius Vach, Christian Rubbert, David Latz, Gerald Antoch, Hans-Jörg Wittsack, Sven Nebelung, Lena Marie Wilms

**Affiliations:** 1Department of Diagnostic and Interventional Radiology, Medical Faculty and University Hospital Düsseldorf, D-40225 Düsseldorf, Germany; anja.lutz@med.uni-duesseldorf.de (A.M.-L.); danielbenjamin.abrar@med.uni-duesseldorf.de (D.B.A.); marius.vach@med.uni-duesseldorf.de (M.V.); christian.rubbert@med.uni-duesseldorf.de (C.R.); antoch@med.uni-duesseldorf.de (G.A.); hans-joerg.wittsack@med.uni-duesseldorf.de (H.-J.W.); snebelung@ukaachen.de (S.N.); lena.wilms@med.uni-duesseldorf.de (L.M.W.); 2Department of Orthopedic and Trauma Surgery, University Hospital, D-40225 Düsseldorf, Germany; david.latz@med.uni-duesseldorf.de; 3Department of Diagnostic and Interventional Radiology, University Hospital Aachen, D-52074 Aachen, Germany

**Keywords:** dens axis fracture, CT imaging, deep learning, radiomics, machine learning

## Abstract

**Background/Objectives**: The present study investigates the influence of advanced windowing techniques and the combination of different classification methods on the accuracy of dens axis fracture detection in computed tomography (CT) images. The aim was to evaluate and compare the diagnostic performance of two different computational models—a pure deep learning (DL) approach and a combined approach of DL segmentation, windowing, and radiomics. **Methods**: In this retrospective study, CT datasets of the upper cervical spine of 366 patients were included. All datasets were further divided into training, validation, and test sets. Model 1 (M1) relied on a pure DL method using a Convolutional Neural Network (CNN) and a Feedforward Neural Network (FNN), without prior manual segmentation. Model 2 (M2) incorporated a fully automatic U-Net-based segmentation followed by radiomics feature extraction and classification using a Machine Learning (ML) Classifier. The performance of both models was measured by classification accuracy, with a particular focus on the impact of CT windowing parameters and the chosen ML classification strategies. **Results**: M1 achieved a maximum classification accuracy of 93.7%, while M2 accomplished a classification accuracy of up to 95.7% by using ROI-based windowing and advanced feature extraction. **Conclusions**: Integrating advanced windowing techniques, U-Net segmentation, and radiomics improves the detection of dens axis fractures in CT imaging. This approach could enhance diagnostic accuracy and warrants further exploration and clinical integration.

## 1. Introduction

Computed tomography (CT) imaging remains a cornerstone of diagnostic precision in medical practice, particularly in the detection of bony injuries such as dens axis fractures. The dens axis, a critical structure within the cervical spine, presents a diagnostic challenge in cases where fractures are subtle. If missed, these can lead to serious clinical implications including instability or worse neurological deficits, due to its proximity to the upper cervical spinal cord and lower brainstem [1]. Therefore, enhancing image processing and visualization techniques is essential to facilitate accurate diagnosis and timely clinical intervention.

In recent years, advancements in radiological imaging and computational techniques have significantly improved diagnostic accuracy in medical imaging [2,3]. AI-based methodologies, specifically deep learning (DL), have proven highly effective for automatic segmentation and classification tasks in medical images, achieving robust results across various imaging modalities and clinical applications [4,5,6].

However, despite impressive capabilities, DL-only approaches cannot fully exploit the complex and nuanced information inherent in medical images, particularly when it comes to detecting subtle pathologies.

Radiomics, a technique involving the extraction of high-dimensional quantitative features from medical images, has emerged as a complementary strategy to DL, enhancing interpretability and diagnostic accuracy [7,8,9,10,11,12]. Radiomics has shown substantial promise across multiple diagnostic domains, including oncology, neurology, and musculoskeletal disorders, by capturing intricate texture-, morphology-, and intensity-based patterns not readily discernible by visual inspection alone [8,9,12].

Recent work in bladder cancer has further demonstrated that radiomics can non-invasively differentiate tumor grade and stage with clinically meaningful accuracy, even in small patient cohorts [13].

Recent studies emphasize the advantages of combining segmentation-driven approaches with radiomic analysis to improve diagnostic outcomes [7,8,9,10,11,12]. For example, Liang et al. demonstrated enhanced accuracy in breast cancer classification through integrating radiomics with segmentation [14], while Gutsche et al. illustrated superior predictive performance for brain metastases using radiomics compared to semantic features alone [15]. Such findings underscore the potential benefit of combining precise anatomical delineation through segmentation with detailed radiomic analysis.

Thus, in this study, we propose an integrated analytical pipeline that combines sophisticated CT windowing techniques, advanced convolutional neural network-based segmentation, and radiomics-driven classification to enhance the detection accuracy of dens axis fractures in a diverse CT dataset. This approach capitalizes on the strengths of both DL and radiomics, leveraging advanced image processing strategies—including Contrast-Limited Adaptive Histogram Equalization (CLAHE), histogram-based windowing, and region-of-interest (ROI)-based windowing—to optimize image visualization and fracture delineation [16,17,18,19]. Our objective is to systematically evaluate the effectiveness of this integrated pipeline compared to traditional DL-only methods, assessing its robustness, accuracy, and clinical applicability with different image processing methods and classification algorithms. Our hypothesis is that this integrated approach will significantly outperform traditional DL-only methods in terms of diagnostic accuracy and robustness.

## 2. Materials and Methods

### 2.1. Study Design and Patient Selection

This retrospective comparative imaging study was conducted in accordance with stringent local ethical guidelines and approved by the Ethics Committee of the Faculty of Medicine at Heinrich Heine University, Düsseldorf, Germany (study reference number: 2020-850).

A total of 400 CT scans of the upper cervical spine, acquired at the University Hospital Düsseldorf between October 2016 and March 2020, were reviewed. These scans were retrieved using various search terms (i.e., “upper cervical spine”, “CT skull + cervical spine”, and “cervical spine”) to ensure a deliberate diversity of imaging perspectives. As a result, the dataset included a wide range of CT scans—from those solely depicting the upper cervical spine to those encompassing both part of the skull and the whole cervical spine—thereby increasing the complexity and realism of the data and highlighting challenges in accurate segmentation and analysis. Each scan was evaluated by an experienced clinical radiologist (L.M.W., with seven years of experience in musculoskeletal imaging) to confirm complete visualization of the dens axis. Exclusion criteria included postoperative CT scans, fractures other than dens axis fractures, other pathological findings, incomplete or truncated imaging, and pediatric CT examinations.

Following the application of these criteria, 366 CT scans were selected for further analysis, comprising 287 without dens fractures and 79 with dens fractures. The dataset was then randomly divided into training (70%, *n* = 256), validation (10%, *n* = 37), and test (20%, *n* = 73) sets, ensuring a systematic and balanced approach to model development and evaluation procedure.

### 2.2. Image Data Acquisition and Data Processing

All CT scans were acquired as part of routine clinical practice at the University Hospital Düsseldorf, Germany. To ensure a representative variety of imaging conditions, three different Siemens Healthineers CT scanners were utilized: SOMATOM Definition Flash (*n* = 245); SOMATOM Definition AS (*n* = 113); SOMATOM Definition Edge (*n* = 8).

The scans exhibited an average slice thickness of 1.94 ± 0.23 mm (1–2 mm), with a mean in-plane pixel resolution of 0.55 ± 0.16 mm (0.17–0.98 mm). The computed tomography dose index volume (CTDI_vol) averaged 13.19 ± 1.91 mGy (3.10–31.12 mGy).

#### 2.2.1. Manual Segmentations

Manual segmentation of the dens and, when applicable, the associated fractures, was conducted on axial CT slices using ITK-SNAP (v3.8, Cognitica, Philadelphia, PA, USA) [20]. Segmentations were performed by L.M.W., who delineated each axial slice containing the dens to create reference ground truth segmentations. Following initial segmentation, all masks underwent visual quality control to ensure that only pixels unequivocally belonging to the dens were retained, thereby reducing partial volume effects (Figure 1).

#### 2.2.2. Data Pre-Processing and Contrast Adjustment

To standardize the dataset, all CT images and corresponding segmentation masks were resampled to an isotropic resolution of 1 × 1 × 1 mm. Linear interpolation was used for CT images, while nearest-neighbor interpolation was applied to the segmentation masks to preserve label integrity. Following this spatial normalization, a range of contrast adjustment and windowing optimization techniques was applied to enhance image quality and create a comprehensive dataset for further analysis (Figure 2).

**Bone Windowing:** A conventional windowing technique that enhances bone visualization by using a standard clinical window setting of 400 Hounsfield units (HU) for the window level and 2000 HU for the window width [21]. This setting highlights bone density and structure, facilitating better identification and assessment of bone pathologies.**Bone Gamma Correction:** This post-processing technique modifies the intensity distribution of bone-windowed images by applying gamma correction. Two gamma values were tested: γ = 0.5 to enhance brightness and γ = 2.0 to reduce it [16]. Gamma values less than 1 emphasize darker areas, while values greater than 1 reduce overall brightness, allowing finer details in both low- and high-intensity regions to emerge, particularly aiding in the detection of bone fractures.**Histogram-Based Windowing:** In this method, the window boundaries are determined by analyzing the histogram of the entire image. The 5th and 95th percentile values of the image’s pixel intensity data are used to set the window boundaries, effectively filtering out extreme pixel values [22]. This technique optimizes the contrast distribution across the entire image, reducing overexposure and underexposure and thereby improving image quality.**Contrast-Limited Adaptive Histogram Equalization (CLAHE):** CLAHE increases local contrast by dividing the image into smaller segments and applying histogram equalization to each segment individually [17,18]. This method improves the visibility of subtle structures, especially in areas of low contrast, without excessively increasing global contrast. While commonly applied in lung imaging [18,19], recent evidence supports its utility in bone imaging—for example, Park et al. demonstrated that CLAHE improved classification performance in scintigraphy [19].**ROI-Based Windowing:** This targeted method utilizes the segmentation mask of the dens region to calculate the 5th and 95th percentile intensity values within the ROI and its immediate surroundings. These values are then applied uniformly to the entire image to maintain consistent contrast and prevent localized discrepancies. This approach ensures clinical readability and, in a two-stage pipeline such as Model M2, can be fully automated using upstream segmentation results.

#### 2.2.3. Data Augmentation

The performance of neural networks is strongly influenced by the size and variability of the training dataset [23]. To enhance model robustness and mitigate overfitting, several data augmentation strategies were applied exclusively to the training set.

The augmentation pipeline began with Z-score normalization, where the mean intensity was subtracted and divided by the standard deviation for each image. This process standardized the dataset to have a mean of zero and a standard deviation of one, thereby stabilizing the training process through a uniform intensity distribution.

Subsequently, all images were resampled and standardized to a uniform voxel resolution of 256 × 256 × 256, with zero-padding applied as needed to maintain anatomical completeness. To introduce variability in anatomical presentation and imaging conditions, random scaling transformations were then performed, resizing the images by a factor ranging from 0.8 to 1.2. Additionally, mirror flipping along the sagittal, coronal, and axial planes was performed to improve the model’s orientation invariance.

Finally, to emphasize the ROI while still preserving contextual variability, the augmented training images were randomly cropped from their 256 × 256 × 256 format to a size of 120 × 120 × 120 voxels. This cropping ensured that at least 90% of the dens axis remained visible in each volume, thereby preserving diagnostic relevance.

Augmentations were applied on-the-fly during training, rather than by generating a permanently augmented dataset. Each patient case was thus presented multiple times per epoch in randomized, slightly altered forms. During training, fracture cases were sampled more frequently (10 times per epoch) compared to non-fracture cases (3 times per epoch), which contributed to balancing the classes and increasing robustness without inflating the dataset size.

In contrast, the validation and test datasets were not subjected to random cropping. Instead, a single, centered 120 × 120 × 120 voxel region was extracted from each standardized 256 × 256 × 256 image volume, centered around the dens. This consistent and standardized cropping approach ensured that all evaluation samples were processed under identical conditions, thereby providing a stable and fair benchmark for model comparison and performance assessment.

### 2.3. CNN-Based Segmentation and Classification

Convolutional neural networks (CNNs) have already demonstrated outstanding performance in medical image analysis, particularly for CT-based segmentation tasks [24,25]. In this study, we compare two distinct DL architectures to fully automate the detection of dens axis fractures from CT images, each leveraging different computational strategies to enhance diagnostic accuracy.

Model 1 (M1) is a single-stage architecture that combines a CNN with a feedforward neural network (FNN). It processes entire CT volumes without predefined ROIs, relying solely on learned features for direct classification into “fracture” or “no fracture” categories.

Model 2 (M2) adopts a two-stage pipeline. First, it employs U-Net-based segmentation to first identify the dens axis from the CT image. In the second stage, radiomic features are extracted from the segmented ROI and used to train a variety of ML classifiers. During training, M2 utilizes manually annotated ROIs to guide the segmentation network. Once trained, it autonomously segments unseen CT scans, enabling a fully automated end-to-end workflow suitable for clinical deployment.

#### 2.3.1. M1—CNN- and FNN-Based Classification

Model M1 is a single-stage architecture that integrates a CNN with a FNNto classify dens axis CT images into binary categories: “with fracture” or “without fracture” (Figure 3, top). The CNN component incorporates DoubleConv modules, each consisting of two consecutive 3 × 3 convolutional layers, followed by batch normalization and ReLU activation functions. Max-pooling layers are inserted between modules to progressively downsample the feature maps while extracting spatial dimensions and hierarchical features. Following feature extraction, the resulting feature maps are flattened and passed through a five-layer FNN. Dropout layers (rate = 0.1) are applied between these layers to reduce overfitting. The final output layer uses a softmax activation function to generate class probabilities, enabling robust binary classification.

#### 2.3.2. M2—CNN-Based Segmentation and Radiomics Analysis

Model M2 incorporates a two-step methodology to improve upon M1 by incorporating a dedicated segmentation stage prior to classification (Figure 3, bottom). In the first step, a 3D U-Net architecture segments the dens region from the CT image. This encoder–decoder structure with skip connections captures both global context and fine-grained details, ensuring accurate ROI isolation for subsequent analysis.

In the second step, radiomics features are extracted from the segmented dens region using the PyRadiomics framework [26]. These features include quantitative descriptors such as texture patterns (e.g., gray-level co-occurrence matrix features), shape metrics (e.g., surface area, volume), and intensity statistics (e.g., first-order histogram features). This comprehensive feature set encapsulates the complex characteristics of the dens that may indicate the presence of fractures. The extracted radiomics features are then input into multiple machine learning classifiers—decision trees, extra trees, gradient boosting machines, k-nearest neighbors (KNN), Gaussian Naive Bayes, and random forests. By evaluating these diverse algorithms, we identify the most effective classifier, leveraging the strengths of ensemble and distance-based methods to manage the high-dimensional radiomic feature space.

#### 2.3.3. Computational Implementation and Evaluation

All models were trained on a high-performance workstation equipped with two Intel Xeon Gold 6242R CPUs (Intel Corporation, Santa Clara, CA, USA), 376 GB RAM, and four NVIDIA RTX 3090 GPUs (NVIDIA, Santa Clara, CA, USA). Model development was performed in PyTorch (v2.3.1) [27], with PyTorch Lightning (v2.3.2) used to facilitate streamlined and efficient training procedures [28].

The Adam optimizer was employed with an initial learning rate of 0.001 for M1 and 0.01 for M2 to accommodate the differing complexities of the two models. A weight decay of 10^−6^ was applied to mitigate overfitting [29]. Learning rates were dynamically adjusted using a “ReduceLROnPlateau” scheduler, configured with a patience of 10 epochs, a cooldown period of 10 epochs, and a reduction factor of 0.1. Training was conducted for a maximum of 50 epochs with a batch size of four, using input volumes standardized to 120 × 120 × 120 voxels.

For the segmentation tasks in M2, a combined Dice and cross-entropy loss was applied [30] to address class imbalance and improve segmentation accuracy. For classification tasks in both M1 and M2, standard cross-entropy loss was utilized [31]. To compensate for the unequal distribution of fracture and non-fracture cases, class weights were introduced into the loss function based on the observed prevalence in the dataset. This weighting scheme ensured that misclassified fracture cases were penalized more strongly, thereby maintaining sensitivity despite the lower number of positive cases. In combination with the on-the-fly augmentation strategy, this approach provided a balanced optimization during training. The model achieving the best performance on the validation dataset was selected for final evaluation on the independent test set.

### 2.4. Evaluation Metrics and Statistical Analysis

Model performance was evaluated using distinct metrics tailored to the two key tasks of this study: segmentation and classification. For segmentation in M2, the Dice Similarity Coefficient (DSC) was used to quantify the spatial overlap between automatically generated segmentations and the manually annotated ground truth. Ranging from 0 (no overlap) to 1 (perfect overlap), the DSC quantifies directly reflect the model’s ability to accurately localize the dens—a prerequisite for reliable radiomic analysis.

For classification performance, accuracy was used as the principal metric. Defined as the proportion of correctly classified cases (fractured vs. no fracture) out of the total number of cases, accuracy provides an intuitive and interpretable measure of diagnostic performance.

To further investigate the dependence of radiomics-based classification on segmentation quality, a stability analysis was performed. Radiomics feature extraction was conducted after segmentation, and the DSC was systematically varied within a controlled range. This procedure enabled assessment of how deviations from perfect segmentation impacted classification performance across different machine learning classifiers, as segmentation methodology has previously been shown to influence radiomics feature stability and reproducibility in imaging studies [32].

Additionally, descriptive statistical methods—including the use of heat maps—were used to visualize model performance across different windowing techniques, ML classifiers, and segmentation accuracies. These visual tools facilitated intuitive interpretation of interdependencies among model components and reinforced confidence in the system’s reliability and clinical applicability.

## 3. Results

### 3.1. Segmentation Performance

M2’s U-Net-based segmentation demonstrated consistently high precision, with DSC values ranging from 0.82 to 0.94 across the test dataset. As summarized in Table 1, the histogram-based windowing technique yielded the highest mean DSC (0.94 ± 0.01), closely followed by CLAHE (0.92 ± 0.01). Both techniques significantly enhanced local contrast around the dens axis, facilitating more accurate delineation of this critical anatomical structure.

**Figure 4 diagnostics-15-02599-f004:**
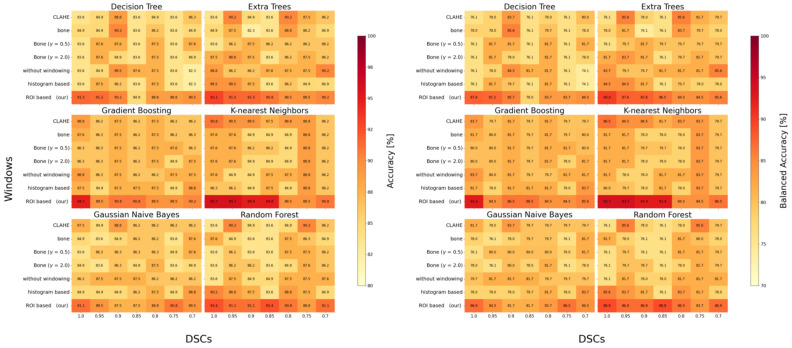
Heatmap illustrating classification accuracy (**left**) and balanced accuracy (**right**) of Model 2 (M2) across varying Dice Similarity Coefficient (DSC) levels, windowing techniques, and machine learning classifiers. Each cell represents the classification performance under a specific combination of segmentation quality and preprocessing method. Darker shades indicate higher accuracy. The visualization highlights M2’s robustness and adaptability, demonstrating consistent performance across multiple classifiers and preprocessing strategies—even when segmentation quality varies. This stability analysis emphasizes the importance of verifying segmentation masks when applying such a two-stage pipeline in clinical routine. Abbreviations: CLAHE—Contrast Limited Adaptive Histogram Equalization; Bone—windowed bone image; γ—gamma adjustment parameter; DSC—Dice Similarity Coefficient.

### 3.2. Classification Performance

**M1 Performance:** M1, which relies solely on a CNN-FNNpipeline without predefined segmentation, achieved a peak classification accuracy of 93.7% on unprocessed images (Table 1). Applying ROI-based windowing marginally improved this result to 94.2%, while other preprocessing techniques showed limited added benefit. The corresponding balanced accuracy values were consistently lower (73.4–88.1%), indicating that M1 was more affected by the underlying class imbalance and tended to favor the majority class.

**M2 Performance:** In contrast, M2 consistently outperformed M1, achieving classification accuracies above 95% across all windowing methods tested. The highest accuracy—95.7%—was obtained using ROI-based windowing combined with a KNN classifier. As illustrated in Figure 4 (left), M2 maintained high classification performance across multiple ML classifiers—including Decision Trees, Extra Trees, Gradient Boosting, KNN, Gaussian Naive Bayes, and Random Forest. ROI-based windowing consistently emerged as the most effective preprocessing method, enabling detection of even minimally displaced fractures. A dedicated stability analysis revealed that classification accuracy declined by less than 1% as the DSC values dropped from 0.94 to 0.82. Furthermore, Figure 4 (right) highlights the stability of M2 K-nearest Neighbors classification across segmentation conditions. Importantly here is that the difference between accuracy and balanced accuracy remained marginal (<2%), which further supporting that M2 maintains a high true-positive rate despite the unequal class distribution. It is important to emphasize that in settings with lower overall accuracy, the corresponding balanced accuracy declined even more markedly. This indicates that, for these specific windowing–classifier combinations, the primary source of performance loss was a reduced sensitivity, i.e., a considerable number of fractures would have been missed. This finding underlines the clinical relevance of classifier choice, as inappropriate configurations may compromise the detection of subtle but critical fractures despite seemingly acceptable overall accuracy values.

## 4. Discussion

This study demonstrates that a two-stage approach—combining U-Net-based segmentation with radiomics-driven classification and targeted windowing adaptations—significantly improves the detection of dens axis fractures in CT imaging within the context of our dataset. Compared to a conventional single-stage DL-based model (M1), the integrated pipeline (M2) consistently achieved higher classification accuracies, reaching up to 95.7%, while maintaining robust performance across multiple preprocessing techniques and ML classifiers. Importantly, this performance was accomplished even with segmentation DSC values ranging from 0.82 to 0.94, indicating the model’s strong resilience to imperfect ROI delineation. Such results highlight the potential of blending advanced segmentation, feature extraction, and ML classification for addressing complex radiological tasks involving subtle anatomical changes.

Our findings support emerging evidence that pure one-stage DL strategies, though powerful, may not fully leverage the complexity of medical imaging data—particularly in identifying subtle pathologies [33,34]. Dens axis fractures exemplify such diagnostic challenges, as they might be morphologically inconspicuous and can easily be overlooked. By isolating the ROI and extracting interpretable, high-dimensional radiomic features, M2 effectively capitalizes on subtle textural, shape, and intensity patterns. This synergy between anatomical localization and radiomic analysis improves the detection of even minimally displaced fractures—an area where end-to-end DL-only models like M1 may underperform due to limited spatial specificity. This advantage resonates with observations from other domains. Our results corroborate the added value of segmentation-driven radiomics in other contexts [14,15] and extend these findings to dens-axis fracture detection. While our study focuses on fracture detection, this approach is broadly applicable to other diagnostic and prognostic imaging tasks.

To place our findings into context, Table 2 provides a comparative summary of previously published studies that investigated AI-based methods for fracture detection in the cervical and vertebral column using CT imaging. Most prior work has concentrated on the entire cervical spine or on thoracolumbar levels, where fractures tend to be larger and therefore more conspicuous. Reported accuracies range between 70% and 98%, depending on dataset size, class balance, and methodological design. Importantly, none of these studies has specifically focused on dens axis fractures. Our work therefore represents the first dens-specific study in this field, highlighting the importance of developing dedicated models for small but clinically critical anatomical structures. Given the potentially life-threatening consequences of missed dens fractures, such tailored approaches may provide substantial added value in clinical decision-making.

Beyond the methodological comparisons, our findings also carry important clinical implications. Missed dens fractures can lead to severe cervical instability and potentially life-threatening neurological complications. An automated pipeline with high diagnostic accuracy could therefore provide valuable support in emergency and trauma settings, where rapid and reliable fracture detection is critical. When used as a second reader or triage system, such tools may help reduce the rate of missed fractures, accelerate decision-making, and enhance efficiency in high-throughput radiology departments. While the present study represents a proof of concept, our results underscore the value of developing dedicated artificial intelligence (AI) solutions for small but clinically decisive anatomical structures such as the dens. Prospective multicenter studies will be required to confirm these benefits in real-world workflows and to quantify their impact on patient outcomes.

In addition to model architecture, our findings underscore the role of targeted contrast enhancement techniques, such as ROI-based and histogram-based windowing, which enhanced segmentation performance by improving local contrast around the dens axis itself, thereby improving diagnostic accuracy as reflected by higher DSC scores and lower standard deviations. Although global contrast enhancement methods like CLAHE are well-established to improve visibility of subtle features [18], their incremental benefit was limited in this context—possibly due to the already optimized internal representation of CNN-based feature extractors. Nonetheless, contrast techniques that enhance local contrast were associated with improvements in both segmentation precision and classification accuracy. This targeted strategy may also be adaptable to other clinical scenarios requiring focused anatomical analysis.

Moreover, M2 demonstrated high performance stability across a range of segmentation qualities, underscoring its clinical applicability. A decline in DSC from 0.94 to 0.82 resulted in less than a 1% decrease in classification accuracy, suggesting that M2 does not rely on perfect segmentations. Rather, its diagnostic strength lies in the integration of reasonably accurate anatomical localization with high-dimensional robust radiomic feature analysis. This resilience is particularly relevant in real-world clinical settings, where image quality may vary due to factors such as heterogeneous imaging protocols, patient motion, and scanner settings—a variability also reflected in our study’s deliberately heterogeneous population. These findings indicate M2’s strong potential for clinical translation, pending prospective validation.

The flexibility of M2’s radiomics-based classification stage is another important strength. While the KNN classifier yielded the highest accuracy in our dataset, M2 also performed well across various ML algorithms, including decision trees, random forests, and gradient boosting machines. This adaptability suggests that M2 is well-suited for further optimization and customization for various clinical use cases. Nonetheless, no single classifier is guaranteed to universally outperform others and future investigations may explore ensemble strategies, advanced boosting methods, or hybrid DL-ML architectures [41,42,43], to further optimize classification performance for specific clinical needs, patient populations, or imaging conditions.

Despite these promising results, several limitations must be acknowledged. First, this study should primarily be regarded as a proof-of-concept. Accordingly, no evaluation of runtime feasibility—including inference speed, PACS integration, or workflow impact on radiologists—was performed. Given the retrospective nature and limited size of the dataset, such an assessment was beyond the scope of this work. Nevertheless, clinical translation will ultimately depend not only on accuracy but also on seamless integration into daily radiological workflows and compliance with regulatory frameworks. Second, all data were derived from a single institution and acquired on scanners from the same vendor. Although we deliberately included examinations from different CT systems of one manufacturer to increase heterogeneity, this still constitutes a single-center, single-vendor setting. Such homogeneity limits generalizability to other populations, imaging protocols, and equipment. Variability in scanner hardware, reconstruction algorithms and acquisition parameters is well known to substantially influence radiomics features and thus model performance and remains one of the major challenges for translating AI into clinical routine. To address this, future work should incorporate multi-center, multi-vendor datasets with harmonized acquisition protocols to rigorously evaluate robustness and external validity [44]. Third, while a range of ML classifiers was tested, our selection deliberately focused on comparatively simple but well-established algorithms, reflecting the study’s proof-of-concept character and the limited dataset size. The rationale was to illustrate that ML-based classifiers can still outperform complex DL-only approaches in settings with small sample sizes and subtle imaging patterns. Nevertheless, future work on larger, multi-center datasets should systematically include additional ML methods such as XGBoost or LightGBM to further optimize performance and evaluate robustness across institutions. Fourth, while radiomics features were extracted, no dedicated feature importance analysis was performed. We deliberately refrained from reporting single feature rankings, as their interpretability outside of a high-dimensional multivariate context is limited and strongly model-dependent. Future multi-center studies should systematically assess feature stability and reproducibility across different scanners and protocols. In such a context, feature importance analyses may not only enhance scientific transparency but also support clinical interpretability, by providing radiologists with concrete radiomics-based markers to contextualize model decisions. Fifth, the dataset exhibited class imbalance, with relatively few positive cases of dens axis fracture. Although this reflects clinical reality, it may reduce sensitivity for rare fracture patterns. Strategies such as data augmentation, oversampling, or cost-sensitive training may help mitigate this limitation and refine the model’s capacity to detect subtle or infrequent fracture patterns. Sixth, the range of classifiers evaluated, while representative, was not exhaustive. Additional testing with advanced ensemble methods, dimensionality reduction techniques, or advanced DL architectures could uncover further performance gains and may yield further improvements in accuracy or interpretability [43]. Seventh, only the validation-optimal model was stored and evaluated, which means that complete epoch-wise accuracy and loss curves are not available. This choice reflects a clinically oriented workflow but limits the possibility to illustrate training dynamics in detail. Finally, although the pipeline demonstrated strong retrospective performance, its clinical utility has not yet been validated prospectively. Future studies should assess how M2 integrates into existing PACS workflows, impacts radiologists’ efficiency, and influences patient outcomes. Establishing practical benefits such as reduced diagnostic delays or improved decision-making will be essential to demonstrating clinical value and encouraging broader adoption.

## 5. Conclusions

In conclusion, our integrated pipeline—combining ROI-based windowing, radiomics-driven feature extraction, and advanced machine learning—significantly improves dens axis fracture detection in CT imaging. The two-stage model (M2) outperformed a conventional CNN-based approach, demonstrating strong accuracy and robustness across various preprocessing techniques and classifiers. While these findings highlight potential for improved diagnostic accuracy, prospective studies on larger datasets are needed to validate the approach and assess its clinical integration.

## Figures and Tables

**Figure 1 diagnostics-15-02599-f001:**
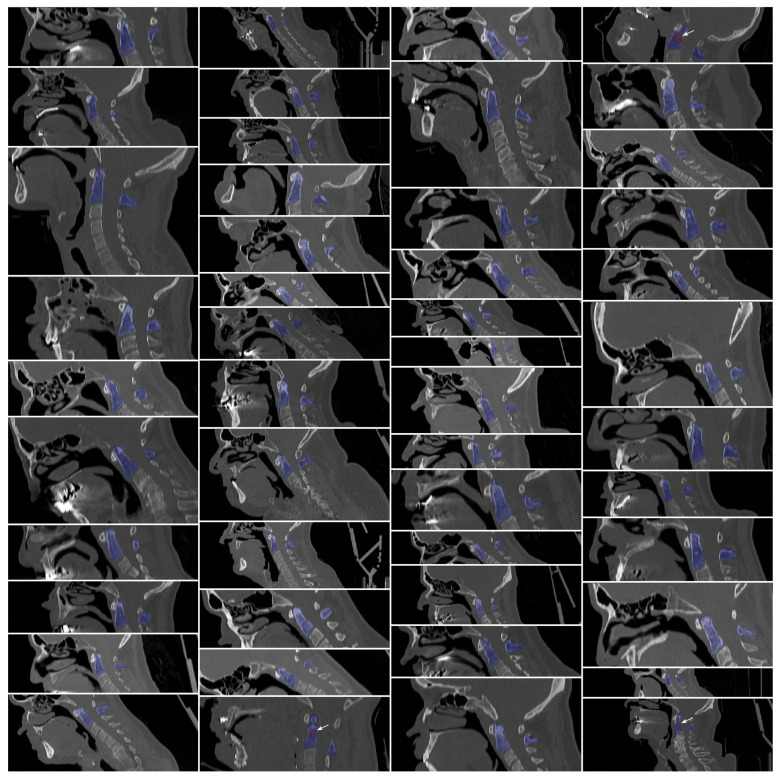
Representative visualization of 48 selected computed tomography (CT) images of the upper cervical spine. The images are sagittally reformatted and display overlaid manual segmentations of the dens axis (blue), and, if present, the corresponding fracture (red). Following manual segmentation, all images were processed according to the data post-processing procedures described in Section 2.2.2 and resampled to an isotropic resolution of 1 × 1 × 1 mm. The selected CT images illustrate the substantial variability within the dataset, including differences in resolutions and anatomical coverage—from scans focused solely on the upper cervical spine to those encompassing the entire skull and cervical spine. This heterogeneity underscores the complexity of the data and highlights the challenges encountered during segmentation, as well as the level of precision required to accurately delineate and analyze these structures.

**Figure 2 diagnostics-15-02599-f002:**
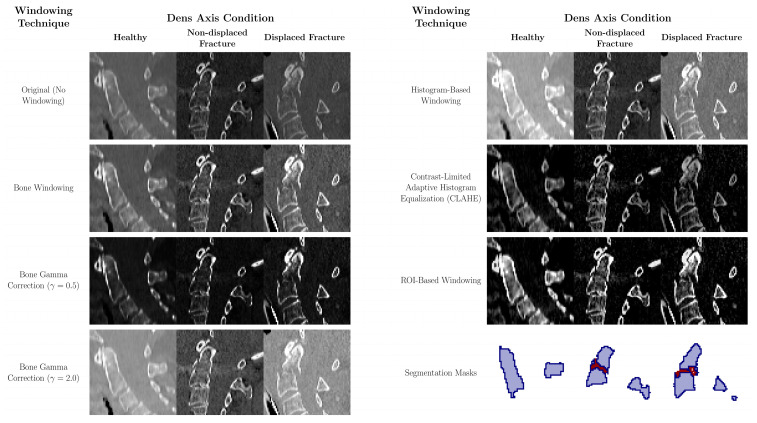
Representative examples illustrating the effects of different contrast adjustment methods on CT images of the dens axis. Shown are three sagittal CT views: one of a healthy dens, one with a non-displaced fracture, and one with a displaced fracture. Each case is presented using both the original contrast and various modified windowing techniques. Of all methods evaluated, ROI-based windowing most effectively enhances fracture visibility. CLAHE and bone windowing also yield noticeable improvements over unadjusted images, though to a lesser extent.

**Figure 3 diagnostics-15-02599-f003:**
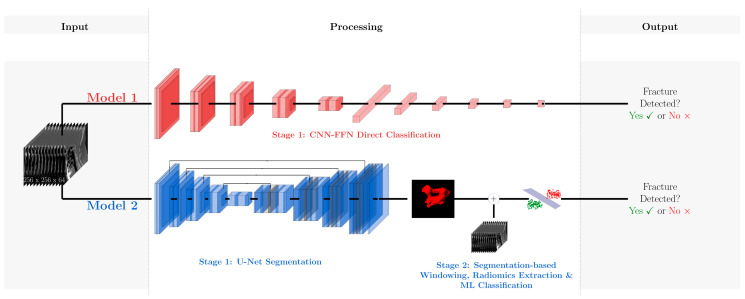
Comparative workflow of Model M1 and Model M2 for dens axis fracture classification. Both models process the same 3D input image. Model M1 (**top**) employs a convolutional neural network and a feedforward neural network (FNN) for direct classification, using DoubleConv modules and max-pooling for feature extraction, followed by multiple FNN layers with dropout to prevent overfitting. In contrast, Model M2 (**bottom**) follows a two-stage approach; a U-Net architecture first segments the dens axis (Stage 1), after which radiomic features are extracted for machine learning-based classification (Stage 2), enabling a more targeted and potentially more accurate detection of fractures.

**Table 1 diagnostics-15-02599-t001:** Comparison of classification accuracy and balanced accuracy for Model 1 (M1) and segmentation performance plus classification accuracy (using a K-nearest neighbor classifier) for Model 2 (M2) across different CT windowing techniques. For extended M2 classification results using additional classifiers, see Figure 4.

Windowing	Model Performance		
	Model 1	Model 2	
	Accuracy/Balanced Accuracy	Segmentation DSC (Mean ± SD)	Accuracy/Balanced Accuracy
without windowing	0.937/0.811	0.91 ± 0.01	0.853/0.865
CLAHE	0.875/0.734	0.92 ± 0.01	0.895/0.817
Bone (γ = 1.0)	0.918/0.834	0.85 ± 0.01	0.849/0.817
Bone (γ = 0.5)	0.932/0.862	0.82 ± 0.03	0.853/0.817
Bone (γ = 2.0)	0.891/0.817	0.90 ± 0.01	0.849/0.817
Histogram-based	0.902/0.825	0.94 ± 0.01	0.862/0.800
ROI-based ^1^	(0.942/0.881)	n/a	0.957/0.937

Abbreviations: M1—Model 1; M2—Model 2; CLAHE—Contrast Limited Adaptive Histogram Equalization; Bone—windowed bone image; γ—gamma adjustment parameter; DSC—Dice Similarity Coefficient; SD—standard deviation. ^1^ ROI-based windowing applies optimal contrast adjustment by calculating the 5th and 95th percentiles of Hounsfield values within a given segmentation. Since this approach requires prior segmentation, no segmentation accuracy was evaluated. The values for M1 were only included to enable comparison with M2. However, ROI-based windowing inherently requires a two-stage approach and is therefore not compatible with the single-stage nature of M1. n/a—Not available because segmentation is necessary for contrast adjustment, which was only generated by the model during segmentation.

**Table 2 diagnostics-15-02599-t002:** Comparative overview of AI-based studies on cervical/vertebral fracture detection in CT.

Study	Region	Preprocessing	Pipeline/Classifier	Performance
Salehinejad et al., 2021 [35]	Cervical fracture (C1–C7)	CT	ResNet50-Cnn + BLSTM	Acc: 70.9–79.2%
Small et al., 2021 [36]	Cervical fracture (C1–C7)	CT	CNN (Aidoc)	Acc: 92%
Chłąd et al., 2023 [37]	Cervical fracture (slice based)	CT, bone window	Yolov5 + Vision Transformer	Acc: 98%
Li et al., 2023 [34]	Occult vertebral fractures (cervical + thoracolumbar)	CT, bone window	Radiomics + ML	Acc: 84.6%
Zhang et al., 2024 [38]	Osteoporotic vertebral fractures (cervical + thoracolumbar)	CT, RadImageNet feature extraction	CNN (RadImageNet vs. ImageNet)	C-Index: 0.795
Singh et al., 2025 [39]	Cervical fracture (C1–C7)	HU normalization	Inception-ResNet-v2 + U-Net decoder	Acc: 98.4%
Liu et al., 2025 [40]	Vertebral fractures + osteoporosis (incl. cervical, thoracolumbar)	CT, RadImageNet pretrained features	CNN (RadImageNet) vs. ImageNet	Acc: 76–80% (per class)
our	Dens axis fracture	ROI-based	U-Net segmentation + Radiomics + KNN	Acc: 95.7%

## Data Availability

Data and evaluation scripts can be provided by the authors upon reasonable request.

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
