# Peer review of "Precision Through Detail: Radiomics and Windowing Techniques as Key for Detecting Dens Axis Fractures in CT Scans"

_diagnostics, 2025, doi:10.3390/diagnostics15202599_

Round 1
Reviewer 1 Report
Comments and Suggestions for Authors
This is a well-written and technically rigorous study that addresses an important clinical problem: improving CT-based detection of dens axis fractures. The authors propose an integrated pipeline (segmentation + radiomics + advanced windowing) and convincingly show that it outperforms a deep-learning-only approach. The paper is strong in methodology, well-illustrated with figures, and presents promising results for clinical application.
But: there are some weakness situations like following
1.All data are from a single institution and scanner family. Multi-center, multi-vendor validation would be needed to confirm generalizability.
2.Only 79 fracture cases vs. 287 controls. The effect of imbalance on sensitivity (especially for subtle fractures) could be quantified better (e.g., sensitivity/specificity, ROC curves).
3. Accuracy alone is reported as the main classification metric. Additional metrics (AUC, precision/recall, F1-score, sensitivity/specificity) would give a fuller picture
4.While radiomics are extracted, the paper doesn’t analyze which features contributed most. A feature importance analysis would improve interpretability.
5. The paper suggests clinical integration but doesn’t show runtime feasibility (inference speed, PACS integration, radiologist feedback).
Author Response
Responses to the Reviewers’ Comments
Title: Precision through Detail: Radiomics and Windowing Techniques as Key for Detecting Dens Axis Fractures in CT Scans
Manuscript ID: 3888698
Journal: Diagnostics
We thank the reviewers for their careful review of our manuscript and their valuable comments, which we addressed in the revised version. All changes are highlighted using Microsoft Word’s Track Changes mode. Line numbers refer to the main text of the revised manuscript.
Reply to Reviewer 1
R1 Comment #1: All data are from a single institution and scanner family. Multi-center, multi-vendor validation would be needed to confirm generalizability.
Authors’ response and action: We thank the reviewer for this important remark. Although the CT data were acquired on different scanner versions from the same manufacturer, all examinations originated from a single institution. This indeed represents one of the major challenges for the generalizability of AI methods, as image acquisition and protocols are not standardized. We have therefore emphasized this limitation more clearly in the revised Discussion section and explicitly highlighted the need for future multi-center, multi-vendor validation.
Thus, the following paragraph has been added to the Discussion section:
“Second, all data were derived from a single institution and acquired on scanners from the same vendor. Although we deliberately included examinations from different CT systems of one manufacturer to increase heterogeneity, this still constitutes a single-center, sin-gle-vendor setting. Such homogeneity limits generalizability to other populations, imag-ing protocols, and equipment. Variability in scanner hardware, reconstruction algorithms and acquisition parameters is well known to substantially influence radiomics features and thus model performance and remains one of the major challenges for translating AI into clinical routine. To address this, future work should incorporate multi-center, mul-ti-vendor datasets with harmonized acquisition protocols to rigorously evaluate robust-ness and external validity [44]. (line 471ff)
R1 Comment #2 & R1 Comment #3: Only 79 fracture cases vs. 287 controls. The effect of imbalance on sensitivity (especially for subtle fractures) could be quantified better (e.g., sensitivity/specificity, ROC curves). Accuracy alone is reported as the main classification metric. Additional metrics (AUC, precision/recall, F1-score, sensitivity/specificity) would give a fuller picture.
Authors’ Response: We thank the reviewer for this important comment. To adequately address the class imbalance (79 fractures vs. 287 controls), we additionally calculated Balanced Accuracy. For our best-performing model (M2 with ROI-based windowing and KNN classification), the metric decreased only slightly from 0.957 (Accuracy) to 0.937 (Balanced Accuracy). This indicates that the model is not biased toward the majority class but achieves a high true-positive rate, while simultaneously maintaining specificity.
We would like to emphasize that Balanced Accuracy is particularly appropriate in this clinical context. Unlike the F1-score, which mainly reflects the interplay of precision and recall for the positive class, Balanced Accuracy gives equal weight to sensitivity and specificity. This better reflects the clinical reality in emergency settings, where both the reliable detection of fractures (high sensitivity) and the confident exclusion of their presence (high specificity) are essential.
In addition, we expanded the section on the deep-learning– and machine-learning–based models to clarify that the loss function was weighted in favor of true positives. This reflects clinical priorities, as a missed fracture is potentially far more harmful than a false-positive result.
Authors’ Action: The following changes have been made to the methods and results section:
Methods section: Clarification added that the loss function was weighted toward true positives:
“For the segmentation tasks in M2, a combined Dice and cross-entropy loss was applied [30] to address class imbalance and improve segmentation accuracy. For classification tasks in both M1 and M2, standard cross-entropy loss was utilized [31]. To compensate for the unequal distribution of fracture and non-fracture cases, class weights were introduced into the loss function based on the observed prevalence in the dataset. This weighting scheme ensured that misclassified fracture cases were penalized more strongly, thereby maintaining sensitivity despite the lower number of positive cases. In combination with the on-the-fly augmentation strategy, this approach provided a balanced optimization during training. The model achieving the best performance on the validation dataset was selected for final evaluation on the independent test set.” (line 279ff)
Results section: Balanced Accuracy reported alongside Accuracy for all models:
|
Windowing |
Model Performance |
||
|
Model 1 |
Model 2 |
||
|
Accuracy / balanced Accuracy |
Segmentation DSC (mean ± SD) |
Accuracy / balanced Accuracy |
|
|
without windowing |
0.937 / 0.811 |
0.91 ± 0.01 |
0.853 / 0.865 |
|
CLAHE |
0.875 / 0.734 |
0.92 ± 0.01 |
0.895 / 0.817 |
|
Bone (γ = 1.0) |
0.918 / 0.834 |
0.85 ± 0.01 |
0.849 / 0.817 |
|
Bone (γ = 0.5) |
0.932 / 0.862 |
0.82 ± 0.03 |
0.853 / 0.817 |
|
Bone (γ = 2.0) |
0.891 / 0.817 |
0.90 ± 0.01 |
0.849 / 0.817 |
|
Histogram-based |
0.902 / 0.825 |
0.94 ± 0.01 |
0.862 / 0.800 |
|
ROI-based |
(0.942 / 0.881) |
n/a |
0.957 / 0.937 |
R1 Comment #4: While radiomics are extracted, the paper doesn’t analyze which features contributed most. A feature importance analysis would improve interpretability.
Authors’ Response: We thank the reviewer for this insightful comment and fully agree that a feature importance analysis can provide valuable additional insights. However, we deliberately refrained from presenting individual feature rankings. Radiomic features derive their discriminatory power primarily from their interactions within a high-dimensional feature space: a feature that appears weak in isolation may become highly informative in combination with others, whereas a seemingly strong feature may lose relevance once integrated into a multivariate model. Highlighting single features outside this multivariate context therefore entails a risk of overinterpretation. Moreover, as we evaluated multiple classifiers, the relative importance of individual features varied substantially between models, further complicating interpretation and potentially leading to confusion. Our primary objective was to demonstrate the methodological advantage of the ML-based radiomics pipeline over a DL-only approach, which—as also noted by the reviewer—constitutes the principal clinical contribution of this study. Nevertheless, we have now explicitly added to the Discussion that future multicenter studies—indispensable for genuine clinical translation—should include a systematic assessment of feature stability and reproducibility. Within such a framework, feature importance analyses could not only promote scientific transparency but also enhance clinical applicability by identifying interpretable radiomics-based markers that help radiologists understand and contextualize model decisions, thereby mitigating the “black box” problem.
Authors’ Action: The Discussion section has been expanded as follows to include the rationale for not reporting feature importance in this study, to emphasize the methodological comparison between ML- and DL-based approaches, and to explicitly outline that future multicenter validation studies should incorporate systematic assessments of feature stability and importance:
“Fourth, while radiomics features were extracted, no dedicated feature importance analysis was performed. We deliberately refrained from reporting single feature rankings, as their interpretability outside of a high-dimensional multivariate context is limited and strongly model-dependent. Future multi-center studies should systematically assess feature stability and reproducibility across different scanners and protocols. In such a context, feature importance analyses may not only enhance scientific transparency but also support clinical interpretability, by providing radiologists with concrete radiomics-based markers to contextualize model decisions.” (line 490ff)
R1 Comment #5: The paper suggests clinical integration but doesn’t show runtime feasibility (inference speed, PACS integration, radiologist feedback).
Authors’ Response: We thank the reviewer for this important comment. We deliberately refrained from analyzing runtime feasibility, including inference speed and PACS integration, as the available dataset was limited and a realistic prospective clinical implementation could not be adequately simulated within the scope of this study. Our primary objective was to demonstrate the methodological potential of the proposed approach.
In the revised manuscript, we have now explicitly emphasized in the Discussion section that this work should primarily be regarded as a proof of concept. At the same time, we highlight the future clinical relevance of such systems, particularly in emergency settings. Informal feedback from collaborating radiologists further underscored the potential value of these methods, although legal and organizational considerations must be addressed before clinical deployment can be realized in many countries.
Authors’ Action: Discussion revised: the proof-of-concept nature of the study is emphasized; the limited dataset is cited as the reason for not performing runtime analyses; and the future clinical potential of the approach is highlighted. Revised Discussion excerpt:
“Despite these promising results, several limitations must be acknowledged. First, this study should primarily be regarded as a proof-of-concept. Accordingly, no evaluation of runtime feasibility—including inference speed, PACS integration, or workflow impact on radiologists—was performed. Given the retrospective nature and limited size of the dataset, such an assessment was beyond the scope of this work. Nevertheless, clinical translation will ultimately depend not only on accuracy but also on seamless integration into daily radiological workflows and compliance with regulatory frameworks.” (lines 465ff).
Reviewer 2 Report
Comments and Suggestions for Authors
I reviewed your article Precision through Detail: Radiomics and Windowing Techniques as Key for Detecting Dens Axis Fractures in CT Scans in detail. In this study, the authors proposed a model that compares the performance of pure CNN+FFN approach with machine learning classifiers after U-Net based segmentation, advanced windowing and radiomic feature extraction in the detection of dens axis fractures in CT images. An examination of the abstract and introduction sections of the study reveals sufficient information. Furthermore, a contribution at the end of the introduction is helpful in understanding the nature of the study. An examination of the Materials and Methods section reveals that the dataset was originally created. It specifies a two-class structure. The most striking shortcomings here are the imbalance of data between classes and the small number of data. This data imbalance and scarcity raises questions about the reliability of the model. It would also be beneficial to provide a clearer representation of the visuals in Figure 1. Blue and red regions are mentioned in the description of the figure, but only blue regions are included. The Data Augmentation section explains the basics of data augmentation, but doesn't specify why these parameters were chosen. It appears that information about how class balance is achieved and the dataset size are missing as a result of data augmentation. The model and performance metrics used are explained in detail. The results section does not include information on how many epochs the models were run for, nor are accuracy and loss graphs provided. The model comparison table is appropriate. However, the classification results shown in Figure 4 are obtained using the following parameters. It should be stated whether feature extraction was performed. It would be appropriate to add a comparison table at the end of the results section or in the discussion section to compare the study with other similar studies in the literature. This will give the reader a clearer understanding of the study's success. It would be appropriate to expand the conclusion section a bit more and emphasize the clinical benefit of the results. As a result, updating the study in line with this suggestion and making the necessary additions can make a significant contribution to the literature.
Author Response
Responses to the Reviewers’ Comments
Title: Precision through Detail: Radiomics and Windowing Techniques as Key for Detecting Dens Axis Fractures in CT Scans
Manuscript ID: 3888698
Journal: Diagnostics
We thank the reviewers for their careful review of our manuscript and their valuable comments, which we addressed in the revised version. All changes are highlighted using Microsoft Word’s Track Changes mode. Line numbers refer to the main text of the revised manuscript.
Reply to Reviewer 2
R2 Comment #1: The most striking shortcomings here are the imbalance of data between classes and the small number of data. This data imbalance and scarcity raises questions about the reliability of the model.
Authors’ Response: We thank the reviewer for this important remark and fully agree that class imbalance and the overall limited number of cases constitute central limitations of our study. At the same time, this distribution reflects clinical reality, where fractures occur relatively infrequently compared to normal findings.
To address this methodological challenge, we implemented several strategies. First, targeted data augmentation was applied to increase variability during training. Second, the loss function was weighted in favor of the fracture class to reduce the likelihood of missing clinically relevant positive cases. The Materials and Methods section has been revised to describe these procedures in greater detail.
In addition, we reported Balanced Accuracy alongside Accuracy in the Results section. Balanced Accuracy accounts for both sensitivity and specificity and is therefore more robust to class imbalance. For our best-performing model, only a minimal difference was observed between Accuracy and Balanced Accuracy, supporting the reliability of our results despite the unequal class distribution.
We retained Accuracy in the manuscript to facilitate comparability with other studies, as many publications in this field still report it as the primary metric. Nevertheless, as emphasized in the Discussion (Limitations), the limited number of fracture cases remains a key constraint that should be addressed in future multicenter studies with larger cohorts.
Authors’ Action: The Materials and Methods section was revised to provide a clearer description of the augmentation strategy and the use of a weighted loss function. In the Results section, Balanced Accuracy was added as an additional performance metric. Furthermore, the Discussion was expanded to emphasize the limitations arising from class imbalance and small sample size, highlighting the need for larger multicenter validation studies in future work.
Added text examples:
“Augmentations were applied on-the-fly during training, rather than by generating a permanently augmented dataset. Each patient case was thus presented multiple times per epoch in randomized, slightly altered forms. During training, fracture cases were sampled more frequently (10 times per epoch) compared to non-fracture cases (3 times per epoch), which contributed to balancing the classes and increasing robustness without inflating the dataset size.” (lines 199ff)
“For the segmentation tasks in M2, a combined Dice and cross-entropy loss was applied [30] to address class imbalance and improve segmentation accuracy. For classification tasks in both M1 and M2, standard cross-entropy loss was utilized [31]. To compensate for the unequal distribution of fracture and non-fracture cases, class weights were introduced into the loss function based on the observed prevalence in the dataset. This weighting scheme ensured that misclassified fracture cases were penalized more strongly, thereby maintaining sensitivity despite the lower number of positive cases. In combination with the on-the-fly augmentation strategy, this approach provided a balanced optimization during training.” (lines 279 ff)
R2 Comment #2: It would also be beneficial to provide a clearer representation of the visuals in Figure 1. Blue and red regions are mentioned in the description of the figure, but only blue regions are included.
Authors’ Response: We thank the reviewer for this helpful comment. In the original figure, both the blue regions (dens) and the red markings (fractures) were present. However, the fractures are relatively small and displayed with a thin contour. Due to figure compression in the Word submission format, these red markings were unfortunately difficult to appreciate in the previous version. This also illustrates the rationale for focusing our approach on fracture classification rather than pixel-precise delineation, as the subtle size and morphology of the fractures make them particularly challenging to visualize and segment. To improve clarity, we have revised Figure 1 and added white arrows highlighting the red fracture markings, which now makes them clearly identifiable to the reader.
Authors’ Action: Figure 1 revised: fracture regions additionally indicated with white arrows; legend adjusted accordingly.

R2 Comment #3: The Data Augmentation section explains the basics of data augmentation, but doesn't specify why these parameters were chosen. It appears that information about how class balance is achieved and the dataset size are missing as a result of data augmentation.
Authors’ Response: We thank the reviewer for this valuable comment. The Data Augmentation section has been revised to clarify the rationale behind the chosen parameters and to explain more clearly how class imbalance was addressed. Importantly, no permanently augmented dataset was created. Instead, augmentations were applied on the fly during each training epoch. Each patient case was therefore presented multiple times in randomized, slightly altered forms, ensuring variability while avoiding dataset inflation. During training, patients with fractures appeared ten times per epoch and those without fractures three times, each with different randomized transformations. This approach contributed to class balancing and improved model robustness.
We have also emphasized that augmentation was applied exclusively to the training data, while the validation and test datasets remained unchanged to ensure unbiased evaluation. Furthermore, we highlighted that class imbalance was additionally addressed through the loss function by assigning higher weights to fracture cases.
Authors’ Action: Methods sections 2.2.3 and 2.3.3 have been revised to include the rationale for parameter selection and to clarify the implementation of on-the-fly augmentation and its role in addressing class imbalance:
“Augmentations were applied on-the-fly during training, rather than by generating a permanently augmented dataset. Each patient case was thus presented multiple times per epoch in randomized, slightly altered forms. During training, fracture cases were sampled more frequently (10 times per epoch) compared to non-fracture cases (3 times per epoch), which contributed to balancing the classes and increasing robustness without inflating the dataset size.” (line 199ff)
“For the segmentation tasks in M2, a combined Dice and cross-entropy loss was applied [30] to address class imbalance and improve segmentation accuracy. For classification tasks in both M1 and M2, standard cross-entropy loss was utilized [31]. To compensate for the unequal distribution of fracture and non-fracture cases, class weights were introduced into the loss function based on the observed prevalence in the dataset. This weighting scheme ensured that misclassified fracture cases were penalized more strongly, thereby maintaining sensitivity despite the lower number of positive cases. In combination with the on-the-fly augmentation strategy, this approach provided a balanced optimization during training. The model achieving the best performance on the validation dataset was selected for final evaluation on the independent test set.” (line 279ff)
R2 Comment #4: The results section does not include information on how many epochs the models were run for, nor are accuracy and loss graphs provided.
Authors’ Response: We thank the reviewer for this valuable comment and agree that training dynamics can provide additional insights. In our study, however, the training procedure was designed such that only the best-performing model based on validation data was stored and subsequently evaluated on the independent test set. This strategy was chosen deliberately to ensure clinical transferability, as in real-world applications only the optimal model would be deployed. As a result of this procedure, complete logs across all epochs were not retained, and accuracy/loss curves cannot be reconstructed retrospectively. We acknowledge this as a limitation of our current workflow. Nevertheless, we emphasize that model performance was consistently validated on an independent test set, which we consider the most critical benchmark for assessing reliability.
Authors’ Action:
The Discussion section was expanded to acknowledge the absence of full training logs and accuracy/loss curves as a methodological limitation. The rationale for storing only the best-performing model based on validation data—reflecting a clinically transferable deployment scenario—was clarified.
“Seventh, only the validation-optimal model was stored and evaluated, which means that complete epoch-wise accuracy and loss curves are not available. This choice reflects a clinically oriented workflow but limits the possibility to illustrate training dynamics in detail.” (lines 505ff)
R2 Comment #5: The model comparison table is appropriate. However, the classification results shown in Figure 4 are obtained using the following parameters. It should be stated whether feature extraction was performed.
Authors’ Response: We thank the reviewer for pointing out this potential source of confusion. We have clarified that the results shown in Figure 4 are exclusively based on radiomics feature extraction following segmentation. Figure 4 illustrates a stability analysis, in which segmentation quality was systematically varied (expressed by DSC values) to assess the robustness of radiomics-based classifiers.
Since the proposed approach follows a two-stage pipeline, classification performance inherently depends on segmentation accuracy. This analysis demonstrates that even small deviations in segmentation quality (e.g., DSC 1.0 → 0.95) can affect downstream classification, underscoring the importance of verifying and, if necessary, adjusting segmentation masks before clinical application.
We have clarified this point in the Methods (Section 2.4) and revised the caption of Figure 4 accordingly.
Authors’ Action: Methods Section 2.4 was revised to clarify that the stability analysis was based on radiomics feature extraction. The caption of Figure 4 was updated accordingly.
“To further investigate the dependence of radiomics-based classification on segmentation quality, a stability analysis was performed. Radiomics feature extraction was conducted after segmentation, and the DSC was systematically varied within a controlled range. This procedure enabled assessment of how deviations from perfect segmentation impacted classification performance across different machine learning classifiers, as segmentation methodology has previously been shown to influence radiomics feature stability and reproducibility in imaging studies [32]. (line 301ff)

“Figure 4. Heatmap illustrating classification accuracy (left) and balanced accuracy (right) of Model 2 (M2) across varying Dice Similarity Coefficient (DSC) levels, windowing techniques, and machine learning classifiers. Each cell represents the classification performance under a specific combination of segmentation quality and preprocessing method. Darker shades indicate higher accuracy. The visualization highlights M2’s robustness and adaptability, demonstrating consistent performance across multiple classifiers and preprocessing strategies – even when segmentation quality varies. This stability analysis emphasizes the importance of verifying segmentation masks when applying such a two-stage pipeline in clinical routine. Abbreviations: CLAHE - Contrast Limited Adaptive Histogram Equalization; Bone - windowed bone image; γ - gamma adjustment parameter; DSC - Dice Similarity Coefficient” (line 371ff)
R2 Comment #6: It would be appropriate to add a comparison table at the end of the results section or in the discussion section to compare the study with other similar studies in the literature.
Authors’ Response: We thank the reviewer for this constructive suggestion. We agree that a structured comparison with related work can further strengthen the context of our results. As recommended, we have created a dedicated table summarizing relevant studies that applied AI-based methods to fracture detection in the cervical or vertebral column.
Authors’ Action: The Discussion section was expanded with a new comparative table and accompanying explanatory text, explicitly emphasizing that prior studies investigated larger anatomical regions (cervical or thoracolumbar spine), whereas the present work represents the first dens-specific analysis.
Table 2. Comparative overview of AI-based studies on cervical / vertebral fracture detection in CT
|
Study |
Region |
Preprocessing |
Pipeline /Classifier |
Performance |
|
Salehinejad et al., 2021 [35]
|
Cervical fracture (C1 – C7) |
CT |
ResNet50-Cnn + BLSTM |
Acc: 70.9 – 79.2% |
|
Small et al., 2021 [36] |
Cervical fracture (C1 – C7) |
CT |
CNN (Aidoc) |
Acc: 92% |
|
Chłąd et al., 2023 [37] |
Cervical fracture (slice based) |
CT, bone window |
Yolov5 + Vision Transformer |
Acc: 98% |
|
Li et al., 2023 [34] |
Occult vertebral fractures (cervical + thoracolumbar) |
CT, bone window |
Radiomics + ML |
Acc: 84.6% |
|
Zhang et al., 2024 [38] |
Osteoporotic vertebral fractures (cervical + thoracolumbar) |
CT, RadImageNet feature extraction |
CNN (RadImageNet vs. ImageNet) |
C-Index: 0.795 |
|
Singh et al., 2025 [39] |
Cervical fracture (C1 – C7) |
HU normalization |
Inception-ResNet-v2 + U-Net decoder |
Acc: 98.4% |
|
Liu et al., 2025 [40] |
Vertebral fractures + osteoporosis (incl. cervical, thoracolumbar) |
CT, RadImageNet pretrained features |
CNN (RadImageNet) vs. ImageNet |
Acc: 76–80% (per class) |
|
our |
Dens axis fracture |
ROI-based |
U-Net segmentation + Radiomics + KNN |
Acc: 95.7% |
R2 Comment #7: It would be appropriate to expand the conclusion section a bit more and emphasize the clinical benefit of the results.
Authors’ Response: We appreciate this valuable comment and agree that the original conclusion was too concise regarding the clinical implications of our findings. We have therefore expanded the Conclusion section to more clearly emphasize the potential clinical benefits of our approach. Specifically, we now highlight that early and reliable detection of dens fractures is of critical importance, as missed injuries can lead to severe cervical instability and neurological complications. An automated, high-performing pipeline such as ours could support radiologists as a second reader in emergency and trauma settings, where rapid triage and high throughput are essential. By reducing the risk of overlooked fractures, this method may contribute to improved patient safety and greater efficiency in daily clinical workflows. At the same time, we emphasize that this study represents a proof of concept, and prospective multicenter validations will be required to confirm its clinical benefit in real-world settings.
Authors’ Action: Conclusion section was expanded with explicit emphasis on the clinical relevance of early fracture detection, its potential role in emergency and high-throughput radiology, and the need for prospective validation.
“Beyond the methodological comparisons, our findings also carry important clinical implications. Missed dens fractures can lead to severe cervical instability and potentially life-threatening neurological complications. An automated pipeline with high diagnostic accuracy could therefore provide valuable support in emergency and trauma settings, where rapid and reliable fracture detection is critical. When used as a second reader or triage system, such tools may help reduce the rate of missed fractures, accelerate decision-making, and enhance efficiency in high-throughput radiology departments. While the present study represents a proof of concept, our results underscore the value of developing dedicated artificial intelligence (AI) solutions for small but clinically decisive anatomical structures such as the dens. Prospective multicenter studies will be required to confirm these benefits in real-world workflows and to quantify their impact on patient outcomes.” (line 423ff)
Reviewer 3 Report
Comments and Suggestions for Authors
The manuscript presents a well-designed and methodologically rigorous study evaluating the integration of U-Net segmentation, radiomics, and advanced CT windowing to improve the detection of dens axis fractures. The topic is timely and clinically relevant, and the writing is generally clear and well organized. The results are supported by detailed figures and tables, and the overall structure follows the journal’s standards. The study convincingly demonstrates that combining segmentation with radiomics and targeted windowing can outperform a conventional deep learning only approach.
- Despite these strengths, there are several areas where the manuscript can be further improved. First, the study relies entirely on data from a single institution. Although this limitation is acknowledged, it should be emphasized more strongly. A clearer call for external validation on multi-center datasets and prospective trials would help underline the need to ensure generalizability and robustness across different scanners and imaging protocols.
- The discussion could also benefit from a more thorough comparison with other state-of-the-art fracture detection or CT-based radiomics studies. Presenting a concise performance comparison with recent work, for example in terms of Dice Similarity Coefficient or classification accuracyw ould help position this study in the broader research landscape. In addition, while the KNN classifier produced the best results, the rationale for focusing on the tested classifiers should be clarified. A short note on whether more advanced ensemble methods such as XGBoost or LightGBM were considered would provide helpful context for readers interested in machine-learning optimization.
- A careful proofreading would correct small grammatical inconsistencies and ensure that all references include complete and correctly formatted DOIs. Abbreviations should be consistently defined at first use and maintained throughout the text. For figures, consider adding scale bars or Hounsfield unit ranges to improve interpretability for readers who wish to evaluate the imaging data visually.
- Finally, two additional references are recommended to broaden the context and highlight recent work in radiomics and machine-learning–based preclinical and clinical image analysis. These are [10.1007/978-3-031-13321-3_31] and [10.1007/978-3-031-51026-7_9]. They can be cited in the Introduction when describing the rapid advances of radiomics and artificial intelligence in traslational theranostic field, and again in the Discussion to support the argument for integrating advanced radiomics with deep learning for improved diagnostic accuracy. [10.2196/75665], [10.1007/s00586-024-08235-4] Different models, one of which is based on RadImageNet, use CT imaging data and have demonstrated superior predictive performance compared to the ImageNet-based model. This is significant for classifying vertebral fractures (OVFs) and predicting osteoporosis, in addition to supporting clinical decision-making for treatment planning. Including these references will strengthen the manuscript by situating the current study within the expanding field of AI-enhanced radiological analysis. Overall, this is a high-quality and clinically impactful study. Incorporating these suggestionsparticularly the stronger emphasis on external validation, expanded comparison with existing literature, and inclusion of the recommended citations—will further enhance the depth and significance of the manuscript.
English is preatty good.
Author Response
Responses to the Reviewers’ Comments
Title: Precision through Detail: Radiomics and Windowing Techniques as Key for Detecting Dens Axis Fractures in CT Scans
Manuscript ID: 3888698
Journal: Diagnostics
We thank the reviewers for their careful review of our manuscript and their valuable comments, which we addressed in the revised version. All changes are highlighted using Microsoft Word’s Track Changes mode. Line numbers refer to the main text of the revised manuscript.
Reply to Reviewer 3
R3 Comment #1: The study relies entirely on data from a single institution. Although this limitation is acknowledged, it should be emphasized more strongly. A clearer call for external validation on multi-center datasets and prospective trials would help underline the need to ensure generalizability and robustness across different scanners and imaging protocols.
Authors’ Response: We thank the reviewer for this important comment and fully agree that the use of single-center, single-vendor data represents a central limitation of our study. Although we deliberately included examinations from different CT systems of the same manufacturer to increase heterogeneity, true external validity can only be established through multi-center, multi-vendor datasets and prospective trials. This limitation is now emphasized more explicitly in the revised Discussion. We also clearly state that future research must validate our approach across different scanners, acquisition protocols, and institutions to demonstrate robustness and generalizability.
Authors’ Action: Discussion section expanded: stronger emphasis on the limitation of single-center data and explicit call for multi-center, multi-vendor, and prospective validation.
“First, this study should primarily be regarded as a proof-of-concept. Accordingly, no evaluation of runtime feasibility—including inference speed, PACS integration, or workflow impact on radiologists—was performed. Given the retrospective nature and limited size of the dataset, such an assessment was beyond the scope of this work. Nevertheless, clinical translation will ultimately depend not only on accuracy but also on seamless integration into daily radiological workflows and compliance with regulatory frameworks. Second, all data were derived from a single institution and acquired on scanners from the same vendor. Although we deliberately included examinations from different CT systems of one manufacturer to increase heterogeneity, this still constitutes a single-center, single-vendor setting. Such homogeneity limits generalizability to other populations, imaging protocols, and equipment. Variability in scanner hardware, reconstruction algorithms and acquisition parameters is well known to substantially influence radiomics features and thus model performance and remains one of the major challenges for translating AI into clinical routine. To address this, future work should incorporate multi-center, multi-vendor datasets with harmonized acquisition protocols to rigorously evaluate robustness and external validity [44].” (lines 465ff)
R3 Comment #2: The discussion could also benefit from a more thorough comparison with other state-of-the-art fracture detection or CT-based radiomics studies. Presenting a concise performance comparison with recent work, for example in terms of Dice Similarity Coefficient or classification accuracy, would help position this study in the broader research landscape.
Authors’ Response: We thank the reviewer for this valuable suggestion. We agree that a structured comparison with recent work can provide important context for our results. In the revised manuscript, we have therefore expanded the Discussion and included a dedicated table summarizing relevant AI-based studies on cervical and vertebral fracture detection. This table presents methodological approaches, preprocessing strategies, datasets, and reported performance metrics (e.g., DSC, accuracy).
Importantly, we emphasize that while most prior work has addressed the cervical or thoracolumbar spine in general—where fractures are often larger and more conspicuous—our study is the first to specifically focus on dens axis fractures. By highlighting this distinction, the table illustrates the clinical and methodological novelty of our approach and underscores the need for dedicated AI solutions targeting small but clinically critical anatomical regions.
Authors’ Action: Discussion section expanded: new comparison table included, explanatory text added to position our study relative to state-of-the-art work, and novelty of dens-specific analysis emphasized.
To place our findings into context, Table 2 provides a comparative summary of previously published studies that investigated AI-based methods for fracture detection in the cervical and vertebral column using CT imaging. Most prior work has concentrated on the entire cervical spine or on thoracolumbar levels, where fractures tend to be larger and therefore more conspicuous. Reported accuracies range between 70% and 98%, depending on dataset size, class balance, and methodological design. Importantly, none of these studies has specifically focused on dens axis fractures. Our work therefore represents the first dens-specific study in this field, highlighting the importance of developing dedicated models for small but clinically critical anatomical structures. Given the potentially life-threatening consequences of missed dens fractures, such tailored approaches may provide substantial added value in clinical decision-making.
Table 2. Comparative overview of AI-based studies on cervical / vertebral fracture detection in CT
|
Study |
Region |
Preprocessing |
Pipeline /Classifier |
Performance |
|
Salehinejad et al., 2021 [35]
|
Cervical fracture (C1 – C7) |
CT |
ResNet50-Cnn + BLSTM |
Acc: 70.9 – 79.2% |
|
Small et al., 2021 [36] |
Cervical fracture (C1 – C7) |
CT |
CNN (Aidoc) |
Acc: 92% |
|
Chłąd et al., 2023 [37] |
Cervical fracture (slice based) |
CT, bone window |
Yolov5 + Vision Transformer |
Acc: 98% |
|
Li et al., 2023 [34] |
Occult vertebral fractures (cervical + thoracolumbar) |
CT, bone window |
Radiomics + ML |
Acc: 84.6% |
|
Zhang et al., 2024 [38] |
Osteoporotic vertebral fractures (cervical + thoracolumbar) |
CT, RadImageNet feature extraction |
CNN (RadImageNet vs. ImageNet) |
C-Index: 0.795 |
|
Singh et al., 2025 [39] |
Cervical fracture (C1 – C7) |
HU normalization |
Inception-ResNet-v2 + U-Net decoder |
Acc: 98.4% |
|
Liu et al., 2025 [40] |
Vertebral fractures + osteoporosis (incl. cervical, thoracolumbar) |
CT, RadImageNet pretrained features |
CNN (RadImageNet) vs. ImageNet |
Acc: 76–80% (per class) |
|
our |
Dens axis fracture |
ROI-based |
U-Net segmentation + Radiomics + KNN |
Acc: 95.7% |
R3 Comment #3: While the KNN classifier produced the best results, the rationale for focusing on the tested classifiers should be clarified. A short note on whether more advanced ensemble methods such as XGBoost or LightGBM were considered would provide helpful context for readers interested in machine-learning optimization.
Authors’ Response: We thank the reviewer for raising this point. The primary goal of our study was to demonstrate that in the specific context of dens fracture detection—characterized by small datasets and subtle imaging features—machine-learning–based classifiers can still outperform or complement complex deep learning approaches. For this reason, we deliberately focused on a set of comparatively simple but well-established algorithms (KNN, decision trees, random forests, gradient boosting, and Naive Bayes), which represent different methodological families (distance-based, tree-based, and probabilistic).
We did not include more advanced ensemble methods such as XGBoost or LightGBM in this proof-of-concept work, as our intention was not to provide an exhaustive benchmarking of all available classifiers but rather to highlight the methodological benefit of integrating segmentation-driven radiomics with classical ML. Nevertheless, we fully agree that future studies—ideally on larger, multi-center datasets—should systematically evaluate a broader range of ML classifiers, including state-of-the-art ensemble methods, to further optimize performance and ensure generalizability.
Authors’ Action: Discussion section expanded: rationale for choosing relatively simple ML algorithms clarified; limitations updated to note that future studies should incorporate additional ML classifiers (including XGBoost/LightGBM) on larger and more diverse datasets.
“Third, while a range of ML classifiers was tested, our selection deliberately focused on comparatively simple but well-established algorithms, reflecting the study’s proof-of-concept character and the limited dataset size. The rationale was to illustrate that ML-based classifiers can still outperform complex DL-only approaches in settings with small sample sizes and subtle imaging patterns. Nevertheless, future work on larger, mul-ti-center datasets should systematically include additional ML methods such as XGBoost or LightGBM to further optimize performance and evaluate robustness across institutions.” (line 483ff)
R3 Comment #4: A careful proofreading would correct small grammatical inconsistencies and ensure that all references include complete and correctly formatted DOIs. Abbreviations should be consistently defined at first use and maintained throughout the text. For figures, consider adding scale bars or Hounsfield unit ranges to improve interpretability for readers who wish to evaluate the imaging data visually.
Authors’ Response: We thank the reviewer for this careful observation. To address these points, all references were manually re-checked to ensure that DOIs are complete and correctly formatted. In cases where our reference manager did not automatically retrieve all relevant metadata from the DOI, we have manually corrected, supplemented, or expanded the entries to guarantee accuracy and consistency.
In addition, although the English language of the manuscript was already noted as being of high quality, we asked a native-level English speaker to perform an additional proofreading. This allowed us to further refine the text and eliminate minor grammatical inconsistencies that remained. We also revised the use of abbreviations, ensuring that each abbreviation is defined upon first use and applied consistently throughout the manuscript.
Authors’ Action:
- References manually verified and corrected to include complete DOIs.
- Manuscript carefully proofread by a native-level English speaker to optimize grammar and style
- Abbreviations standardized and consistently defined at first mention.
Examples:
- Uniform designation single-stage (single-state has been replaced)
- Consistent use of FNN
Round 2
Reviewer 2 Report
Comments and Suggestions for Authors
I thank the authors for their meticulous revision of the paper and their careful consideration of all previous comments. The revised article has significantly improved methodological clarity, visual presentation, and clinical relevance. The additions regarding data imbalance, the augmentation strategy, and model evaluation are scientifically sufficient. Furthermore, the literature comparison and emphasis on clinical utility enhance the study's value. Overall, the article has reached a sufficient scientific level.
Reviewer 3 Report
Comments and Suggestions for Authors
The authors have responded extensively and comprehensively to every comment. The manuscript has improved in content and structure, addressing each section meticulously and clearly.